# The Method of Mass Estimation Considering System Error in Vehicle Longitudinal Dynamics

**Nan Lin [1,2]**, **Changfu Zong [1]** and **Shuming Shi [2,*]**

1   State Key Laboratory of Automotive Simulation and Control, Jilin University, Jilin 130022, China; linnan@jlu.edu.cn (N.L.); Zong.Changfu@jlu.edu.cn (C.Z.)
2   Transportation College, Jilin University, Jilin 130022, China
*   Correspondence: shishuming@jlu.edu.cn; Tel.: +86-15948022008

**Abstract:** Vehicle mass is a critical parameter for economic cruise control. With the development of active control, vehicle mass estimation in real-time situations is becoming notably important. Normal state estimators regard system error as white noise, but many sources of error, such as the accuracy of measured parameters, environment and vehicle motion state, cause system error to become colored noise. This paper presents a mass estimation method that considers system error as colored noise. The system error is considered an unknown parameter that must be estimated. The recursive least squares algorithm with two unknown parameters is used to estimate both vehicle mass and system error. The system error of longitudinal dynamics is analyzed in both qualitative and quantitative aspects. The road tests indicate that the percentage of mass error is 16%, and, if the system error is considered, the percentage of mass error is 7.2%. The precision of mass estimation improves by 8.8%. The accuracy and stability of mass estimation obviously improves with the consideration of system error. The proposed model can offer online mass estimation for intelligent vehicle, especially for heavy-duty vehicle (HDV).

**Keywords:** mass estimation; system error; colored noise; recursive least squares; heavy-duty vehicle

---

## 1. Introduction

Vehicle mass significantly affects the power and efficiency performance. For HDVs, advanced control systems such as automated gear shift strategies [1] and economic cruise control [2] need the information of vehicle mass. For vehicle longitudinal dynamics, acceleration resistance, rolling resistance and grade resistance are all influenced by vehicle mass. Vehicle mass is the most important parameter and potentially has notably large changes [3]. For a HDV, the vehicle total mass can be 10–50 t, with or without a trailer. A variation up to 400% in operation is normal. Hence, it is important to design an online mass estimation method for HDVs.

Generally, mass estimation methods are based on longitudinal dynamics. State estimation algorithms such as the recursive least squares algorithm (RLS) [4] and Kalman filter (KF) [5] are commonly used technologies. Fathy [6] is inspired by perturbation theory and simplifies the mass estimation model with the differential equation of longitudinal dynamics. Similarly, Chu et al. [7] established an electric vehicle (EV) mass estimation model with a high-frequency information extraction method. EVs have advantages in measuring the high-quality tractive force, which guarantees the estimation accuracy. Altmannshofer and Endisch [8] presented a robust parameter algorithm for the vehicle mass and driving resistance estimation. The algorithm overcomes the drawbacks of outliers and insufficient excitation. The error estimation [9] and torque observer [10] are also applied to further improve the accuracy and robustness of the mass estimation.

Many scholars simultaneously estimate the mass and road grade [11–15]. Vahidi [16,17] designed the RLS with multiple forgetting factors to account for different rates of change of different parameters, thus enabling the simultaneous estimation of the time-varying grade and piece-wise constant mass. Lei et al. [18] used an extended KF to estimate both mass and road grade. Although it is notably efficient and convenient to obtain both mass and grade, the disadvantage is also obvious. Since both mass and grade are calculated from longitudinal dynamics, it is difficult to guarantee accuracy when so many variables are left unknown. For other area of moving objects, mass and other structure parameters may be estimated simultaneously. Such as robots [19] and excavator arm [20].

Although the literature has provided many reasonable solutions, good results always rely on precise inputs. Without using laboratory sensors, how to maintain the accuracy and stability remains a challenge.

Too many parameters make it difficult to maintain the precision. The longitudinal dynamics contains approximately 10 constants and 5 variables. Constants such as the roll resistance coefficient are measured in standard test situations. These constants cannot self-adapt to different driving conditions. The most important variable, which is the driving torque, is also calculated by a complicated state estimator [10,21], which may contain many errors. Improving the precision of one or two parameters is not sufficient, and the system error of longitudinal dynamics should be considered.

The system error is always considered a Gaussian distribution in RLS and KF. However, there is a fixed deviation in different driving conditions in longitudinal dynamics. Different fixed deviations can be described as colored noise. Since this fixed deviation represents a tractive or braking force in longitudinal dynamics, we call the system error $F_{se}$. If the system error is mixed with another force in longitudinal dynamics, the accuracy of the mass estimation may decrease.

This study considers the system error as an unknown parameter to identify. The RLS is used to simultaneously identify the vehicle mass and system error. The system error of longitudinal dynamics is qualitatively and quantitatively analyzed. The road test results indicate that the error percentage can decrease by 8.8% by considering the system error.

This paper is organized as follows. In Section 2, the mass estimation method that considers the system error in vehicle longitudinal dynamics is built. Then, the model is verified by field tests in Sections 2 and 3. In Section 4, the system error of longitudinal dynamics is qualitatively and quantitatively analyzed. The off-line fitting data explain how $F_{se}$ causes the lower-precision problem. The road test results indicate that the precision of the mass estimation obviously improves when the system error is considered.

## 2. Mass Estimation That Considers System Error in Vehicle Longitudinal Dynamics

### 2.1. Longitudinal Dynamics with System Error

Mass estimation is based on vehicle longitudinal dynamics, which represents the balance between the tractive force and four types of driving resistance. Shown as Figure 1.

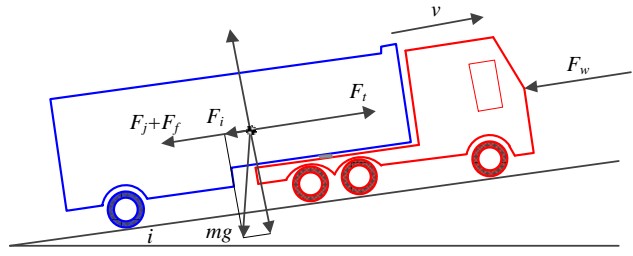

**Figure 1.** Vehicle longitudinal dynamics.

In addition, we express the system error of longitudinal dynamics as follows,

$$F_t = F_f + F_w + F_i + F_j + F_{se} \tag{1}$$

where $F_{se}$ is the system error of longitudinal dynamics. System error represents accumulative errors caused by sensor drift or uncertain environment disturbance in vehicle longitudinal dynamics. It contains the error of both variables and parameters in the tractive force and driving resistance. Different with normal measured noise, such as white noise, system error may change with different driving condition. It is a kind of color noise. In one independent test, its value is considered a constant.

Other tractive forces or resistances are: tractive force $F_t$; roll resistance $F_f$; air drag $F_w$; grade resistance $F_i$; inertia force $F_j$.

For roll resistance,

$$F_f = mgf \tag{2}$$

here $m$ is vehicle mass. $f$ is roll resistance coefficient. $g$ is acceleration of gravity. For air drag,

$$F_w = \frac{1}{2} C_D A \rho v^2 \tag{3}$$

here $C_D$ is air drag coefficient, $A$ is front area and $\rho$ is air density. For tractive force,

$$F_t = \frac{T_{tq} i_0 i_g \eta}{r} \tag{4}$$

here $T_{tq}$ is engine torque. $i_0$ is the final drive. $i_g$ is the transmission. $\eta$ is the efficiency of power train. $r$ is radius of the tire. With the engine speed, vehicle speed and tire radius, the gear ratio of the entire powertrain is,

$$i_0 i_g = 0.377 \frac{rn}{v} \tag{5}$$

here $n$ is the engine speed, which can be obtained from the Controller Area Network (CAN) bus. Bringing (5) into the expression of the tractive force (4), we have a new expression of the tractive force as (6). The tractive force can be calculated without the transmission ratio.

$$F_t = 0.1047 \frac{T_{tq} \eta n}{v} \tag{6}$$

For grade resistance,

$$F_i = mgi \tag{7}$$

here $i$ is the sine of road slope angle. For inertia force,

$$F_j = \delta m a_v \tag{8}$$

$a_v$ is the acceleration of vehicle. $\delta$ is the correction coefficient of the rotating mass, which is illustrated in (9). The correction coefficient of the rotating mass can be expressed as [22],

$$\delta = \left( 1 + \frac{1}{m} \frac{I_w}{r^2} + \frac{1}{m} \frac{I_f i_0^2 i_g^2 \eta}{r^2} \right) \tag{9}$$

where $I_f$ is the rotational inertia of the flywheel, $I_w$ is the rotational inertia of all tires, and the inertial force can be separated into three parts and represented by:

$$F_{ja} = m a_v \tag{10}$$

$$F_{jw} = \frac{I_w}{r^2} a_v \tag{11}$$

$$F_{jf} = \frac{I_f i_0^2 i_g^2 \eta}{r^2} a_v = a_v I_f \eta \left(\frac{0.1047n}{v}\right)^2 \tag{12}$$

Then, Equation (1) becomes

$$F_t = F_f + F_w + F_i + F_{ja} + F_{jw} + F_{jf} + F_{se} \tag{13}$$

According to whether each tractive/resistance is related to the vehicle mass, we arrange them on each side of the equation. On the left side,

$$F_{et} = F_t - F_w - F_{jw} - F_{jf} \tag{14}$$

$F_{et}$ is the equivalent tractive force. This variable represents the tractive force without resistances, which is not related to vehicle mass. On the right side, we have

$$F_f + F_i + F_{ja} + F_{se} = m(gf + gi + a_v) + F_{se} \tag{15}$$

To measure the road grade with high accuracy, a longitudinal acceleration sensor is used [23,24].

$$a_{sen} = gi + a_v \tag{16}$$

$a_{sen}$ is the acceleration measured by the sensor. Since this signal can be measured with high quality and reliability, the system error in this model will not be considered. Its measurement noise is white noise, which can be filtered by the RLS algorithm. Substituting (16) into (15), we can obtain the final form of the mass estimation model,

$$F_{et} = m(gf + a_{sen}) + F_{se} \tag{17}$$

Equation (17) can be used to estimate the vehicle mass and system error.

### 2.2. Recursive Least Squares Model

The RLS model is used to simultaneously identify the vehicle mass and system error. The classic form of RLS with the forgetting factor is used to estimate the vehicle mass, which can be expressed as,

$$Z = \varphi\theta + e_m \tag{18}$$

where $\varphi = (gf + a_{sen} \ 1)$ and $Z = (F_{et})$ are the system input and output, respectively, and $\theta = (m \ F_{se})^T$ is the parameter to be estimated. $e_m$ is the measurement noise. At each time step from $t$ to $t + 1$, the estimation process in discrete form can be expressed as [17],

$$\begin{aligned}
\gamma(t+1) &= L(t)\varphi(t+1)\left[\varphi^T(t+1)L(t)\varphi(t+1) + \lambda\right]^{-1} \\
L(t+1) &= \frac{1}{\lambda}\left[1 - \gamma(t+1)\varphi^T(t+1)\right]L(t) \\
\hat{\theta}(t+1) &= \hat{\theta}(t) + \gamma(t+1)\left[z(t+1) - \varphi^T(t+1)\hat{\theta}(t)\right]
\end{aligned} \tag{19}$$

where $\lambda$ is the forgetting factor, and $L$ is the covariance matrix. The initial value of the covariance matrix is $L_0 = 100$. $L_0$ is selected based on the test data. The value of $\lambda$ can be constant or vary with time. A vary value of $\lambda$ can solve the problem of data saturation. In our research, based on the principle of valid data judgment, much of the data will be filtered. Data saturation is not obvious; thus, we use $\lambda = 1$.

Not all the measured data are qualified for mass estimation. Since the test vehicle cannot offer a braking force, only the acceleration section can be used. The measured data at low speed are generally inaccurate because the gear frequently shifts when the vehicle begins to run. The system input and

output also have boundaries. Based on these reasons, the valid data principle is formed in Table 1. The value of each principle is selected based on the test data.

**Table 1.** Valid data principle.

|  | Principle | Value | Reason |
|---|---|---|---|
| 1 | Minimum speed | >5 m/s | stable running |
| 2 | Clutch engaged | <1 | tractive force output |
| 3 | Braking pedal released | <1 | no braking force |
| 4 | System input saturation (X) | $0.05 \text{ m/s}^2 < X < 0.8 \text{ m/s}^2$ | eliminate abnormal data |
| 5 | System output saturation (Y) | >500 N | eliminate abnormal data |

Data selection will make the RLS method discontinuous. At valid sample times, (17) can be used, whereas, at invalid sample times, the result is only maintained until the next sample time. When the valid data reach 100 s, the mass estimation stops, which may take 200–500 s from the beginning. When the estimation time exceeds 600 s, the calculation is also stopped. The frame of the mass estimation is shown in Figure 2.

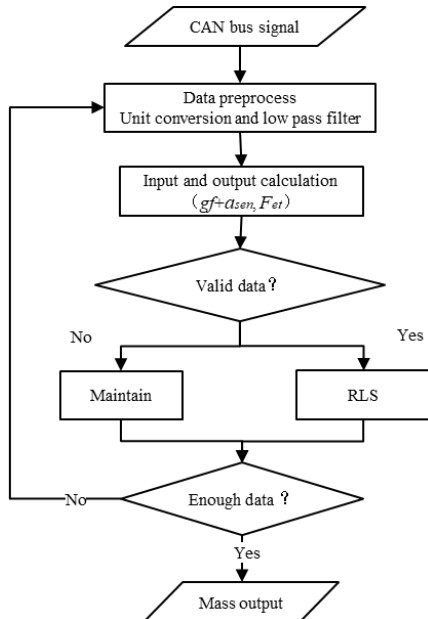

**Figure 2.** Frame of mass estimation.

## 3. Road Test

To verify the constructed model, road tests were performed. The testing ground was a section of an expressway to avoid the controlled environment of a field test. The driver finished his transport task in real traffic conditions.

The test vehicle is a heavy-duty vehicle with a diesel engine and 12-gear transmission. All vehicle parameters in Section 2 are provided by the manufacturer (Table 2). One 1.6-g chassis acceleration sensor was used. It is a kind of Micro Electro Mechanical Systems (MEMS) acceleration sensor. The sensor was installed behind the cab as shown in Figure 3. This position is convenient to install and less affected by the pitching motion. The type of sensor is suitable for production without high cost.

**Table 2.** Parameters in road test.

| Constants | | | |
|---|---|---|---|
| Symbol | Meaning | Unit | Value |
| $\eta$ | efficiency of power train | - | 0.93 |
| $r$ | radius of the tire | m | 0.52 |
| $f$ | roll resistance coefficient | - | 0.0046 |
| $g$ | acceleration of gravity | $m/s^2$ | 9.8 |
| $C_D A\rho$ | air drag coefficient (combined) | - | 10.65 |
| $I_f$ | rotational inertia of the flywheel | $kg \cdot m^2$ | 1.7 |
| $I_w$ | rotational inertia of all tires | $kg \cdot m^2$ | 398.3 |

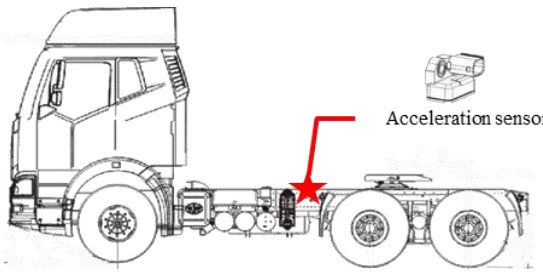

**Figure 3.** Test vehicle and the installation position of the acceleration sensor.

One set of Dewe-43 from Dewetron company is used to simultaneously record the CAN bus signal and sensor signal. The specific channel of measured variables is shown in Table 3. The sample frequency is 20 Hz.

**Table 3.** Channel of measure variables.

| NO | Variables | Unit | Source |
|---|---|---|---|
| 1 | Sample time | s | |
| 2 | Engine speed | rpm | |
| 3 | Engine torque | Nm | |
| 4 | Clutch signal | 0/1 | CAN bus signal |
| 5 | Braking pedal signal | 0/1 | |
| 6 | Vehicle speed | km/h | |
| 7 | Longitudinal acceleration | $m/s^2$ | Acceleration sensor |

Each independent test comprises a complete driving task, which includes starting the vehicle, acceleration, cruising and stopping. The minimum time for each test is 30 min. Three different load tests are performed. The list of road tests is shown in Table 4.

**Table 4.** Road test database with different loads.

| Database | Mass (t) | Number of Tests | Notes |
|---|---|---|---|
| Tractor with full load trailer | 48.0 | 16 | - |
| Tractor with empty trailer | 23.2 | 17 | Winter test |
| Tractor without trailer | 9.5 | 15 | - |

## 4. Mass Estimation Result

### 4.1. Single Test

The time series of road tests are analyzed in this section. Three different loads are illustrated. Each set contains vehicle speed ($v$) system input ($gf + a_{sen}$), output ($F_{et}$), estimated vehicle mass ($m$) and system error ($F_{se}$). The yellow backgrounds indicate those valid sample times.

Figure 4 is the result of a tractor with a full-load trailer. The whole mass estimation algorithm starts when the engine is ignited. The oscillation of measured data at the beginning is obvious, which may be caused by gear shifts. When the speed reaches approximately 20 km/h, the RLS starts to run. The response speed of both $m$ and $F_{se}$ is notably fast. $m$ gradually approaches the real mass, while $F_{se}$ oscillates one or two times. $m$ and $F_{se}$ become stable simultaneously. The algorithm runs for 240 s. The estimated vehicle mass is 48.9 t, and the error is 0.9 t. The estimated system error is −1115 N.

Figure 5 represents one set result of the tractor with an empty trailer. $F_{et}$ significantly decreases. The vehicle mass has significant effects on the engine demand torque. The outliers of $F_{et}$ at the beginning are obvious. The RLS starts until the vehicle speed is 60 km/h. The algorithm runs for 433 s. The estimated vehicle mass is 23.9 t, and the error is 0.7 t. The estimated system error is 1045 N.

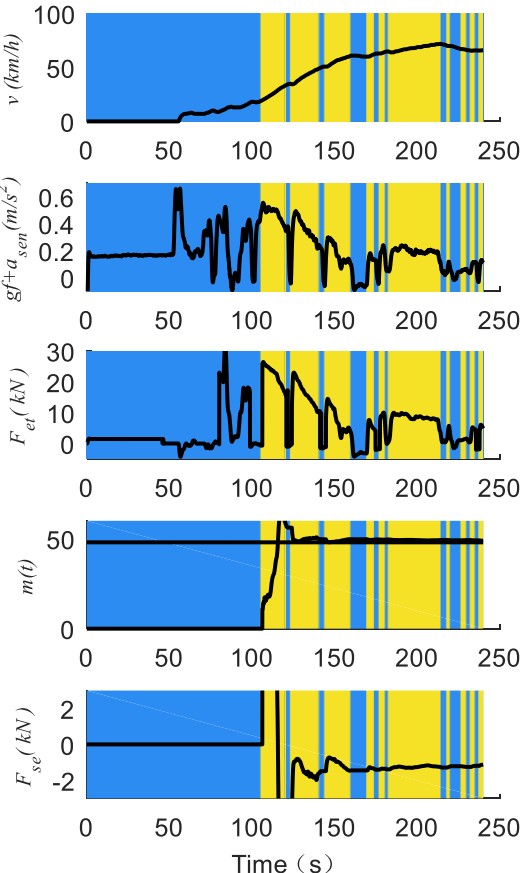

**Figure 4.** One set of the full-load test.

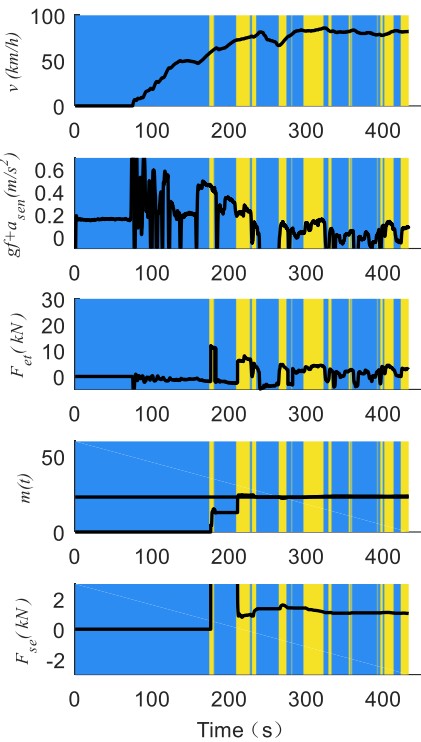

**Figure 5.** One set of the empty trailer test.

Figure 6 is one set of results of the tractor without a trailer. This test contains many deceleration sections. The valid data principle excludes the influence of the deceleration. The discontinuity caused by data selection does not cause any stability problem of the RLS. The algorithm runs for 241 s. The estimated vehicle mass is 9.1 t, and the error is −0.4 t. The estimated system error is −641 N.

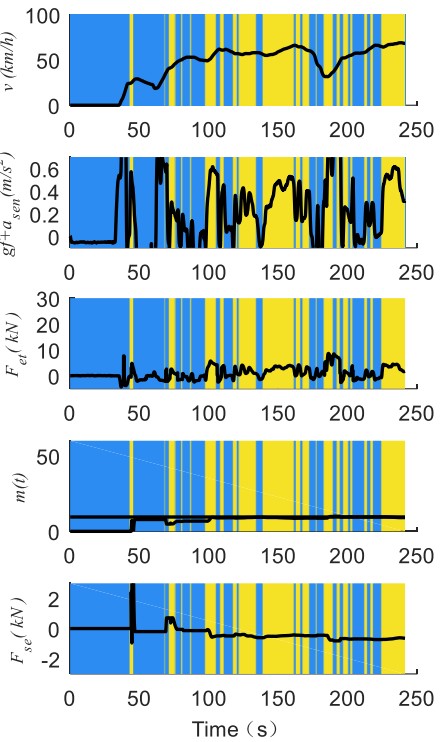

**Figure 6.** One set of the no-trailer test.

## 4.2. Statistical Analysis of Road Tests

All estimation results of the three loads are shown in Figure 7. Each independent test contains the estimated mass, mass error, system error and duration time from the algorithm set up until the termination condition is satisfied. The error between the estimated and real masses is used to analyze the accuracy and stability of the proposed model. The statistical analysis of the mass error is shown in Table 5. The absolute error is used to avoid positive and negative errors canceling each other out.

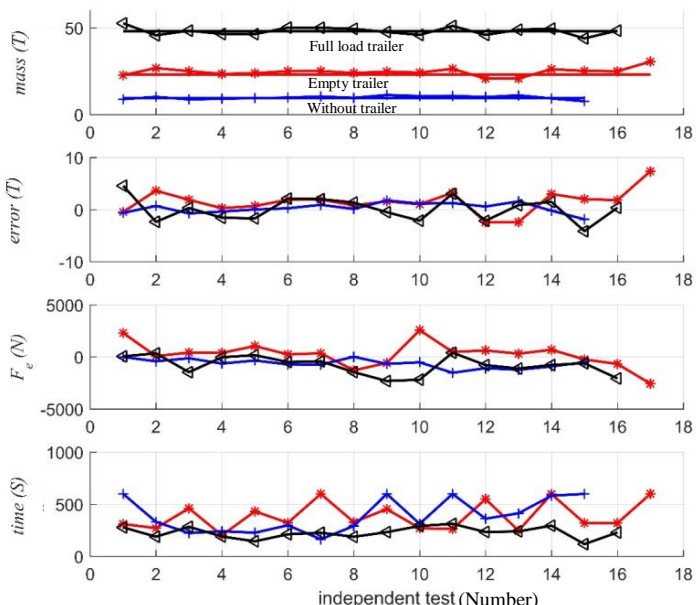

**Figure 7.** Results of the online estimation.

For the estimated vehicle mass, in terms of stability, 98% of the estimated mass error can be controlled within 5 t. The maximum absolute value is 4.6 t. In terms of accuracy, the average absolute value is 1.6 t. The system error has a wide range, from approximately −2000 N to 2000 N, and its value at any specific test is random. Most tests can be finished within 500 s. Full-load tests can finish the estimation in a shorter time because the heavy load reduces the vehicle power performance. The mild acceleration improves the percentage of the valid sample time.

For the full load, its average absolute value is the best (1.9 t), which is only 4% of its real load. Its stability is good, with all estimated errors less than 5 t.

The results of the empty trailer are not notably satisfactory. Its maximum absolute error is 7.3 t, which is greater than that of the other two loads. There is an approximately 94.1% possibility that the error can be constrained to 5 t. The average absolute error is 2.1 t, which is 9.1% of the real mass.

**Table 5.** Statistical analysis of the mass error.

| Tests Condition | Maximum Absolute Value (N) | Percentage within 3 t (%) | Percentage within 5 t (%) | Average | | Average Absolute Value | |
|---|---|---|---|---|---|---|---|
| | | | | Value (t) | Percentage in Real Load (%) | Value (t) | Percentage in Real Load (%) |
| Full-load trailer | 4.6 | 81.3 | 100.0 | 0.1 | 0.2 | 1.9 | 4.0 |
| empty trailer | 7.3 | 82.4 | 94.1 | 1.5 | 6.5 | 2.1 | 9.1 |
| without trailer | 1.9 | 100.0 | 100.0 | 0.3 | 3.2 | 0.8 | 8.5 |
| Total | 4.6 | 87.9 | 98.0 | 0.6 | 3.3 | 1.6 | 7.2 |

The tractor without a trailer has the best stability. The error in all tests is limited to 3 t, with its average absolute error of only 0.8 t. Since its real load is only 9.8 t, the percentage of the average absolute error remains higher than that of the full load. The percentage of the average absolute error of the overall test is approximately 7.2%.

## 5. Analysis of System Error in Vehicle Longitudinal Dynamics

The online estimation results show that the system error considerably varies. Its value is random in each independent test and cannot be ignored. In this section, the formation of the system error will be illustrated in both qualitative and quantitative aspects. Offline fitting and experiments without considering the system error are used for demonstration.

### 5.1. Source of System Error

The foundation of longitudinal dynamics is steady-state force equilibrium. This approach is always used for offline research or analyses such as vehicle performance optimization and vehicle design. Under online driving conditions, the numerical precision of this theoretical force equilibrium is reduced by many situations such as the measured data accuracy, road and weather environment, and vehicle moving status. The meaning of the system error is to self-adapt these disturbances with one parameter. The main source of system error is the following aspect.

### 5.1.1. Accuracy Limitation of Engine Torque from CAN Bus Signal

For traditional internal combustion engines, it is difficult to acquire their output engine torque in online situations. Torque sensors can be used in the laboratory but are not suitable for vehicles on the market. Online torque can be estimated by models such as mean value engine models [21,25]. The characteristic of an engine may change over time; thus, the accuracy of engine torque is difficult to estimate. During online road tests, it is impractical to measure the engine torque of the test vehicle with high accuracy.

This problem can be solved for EVs. The tractive force of EVs can be more easily and accurately measured so that the mass estimation with higher accuracy and stability can be obtained for EVs [7].

### 5.1.2. Constant Parameters

Driving conditions vary considerably, but the vehicle parameters are measured under certain circumstances. It is impossible to make all parameters self-adapt. The mass estimation with one set of vehicle parameters inevitably contains system error.

Most vehicle parameters vary with environmental conditions. The tire radius changes with tire pressure. The roll resistance coefficients change with vehicle speed and road surfaces. For the HDV, the trailer size and the dimensions of transported goods will change the front area and drag coefficient. The total wheel inertia and drag coefficient varies by over 200%.

### 5.1.3. Lateral Motion and Tire Slip

Longitudinal dynamics is based on longitudinal movement. Lateral motion and tire slip are ignored. Since the test vehicle cannot offer the steering wheel angle and lateral acceleration through the CAN bus signal, it is difficult to precisely identify lateral motion. The longitudinal force equilibrium is disturbed while turning.

Figure 8 summarizes the sources of system error. There are too many aspects that may generate system error. To improve the accuracy of mass estimation, the system error without a linear relationship with vehicle mass should be separated from the tractive force. The estimation of system error is a system error self-adapting mechanism, which improves the accuracy and stability of mass estimation.

In addition, for the test HDV, the main source of system error is the engine torque. The percentage of roll resistance is much smaller than other resistances. The inertia from the road slope and acceleration can be measured from sensors, which is highly accurate. The air drag becomes serious when cruising at high speed. The estimation is mostly performed at the beginning of the acceleration; thus, the average speed is not notably high. Based on the results of the test vehicle, the system error of longitudinal dynamics mostly originates from the engine torque.

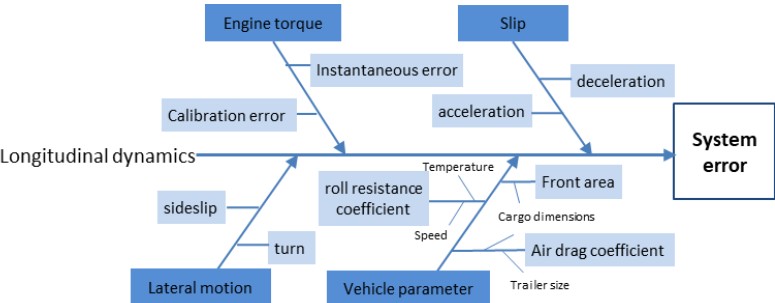

**Figure 8.** Source of system error.

*5.2. Data Fitting in Offline Situation*

In addition, for the test HDV, the main source of system error is the engine torque. The percentage of roll resistance is much smaller than other resistances. The inertia from the road slope and acceleration can be measured from sensors, which is highly accurate. The air drag becomes serious when cruising at high speed. The estimation is mostly performed at the beginning of the acceleration; thus, the average speed is not notably high. Based on the results of the test vehicle, the system error of longitudinal dynamics mostly originates from the engine torque.

The online estimation can be achieved using the recursive least squares algorithm, and it is inconvenient to observe the distribution of all measured data. Only the time series of estimated parameters can be obtained. Offline data fitting is used in this section, which illustrates how the system error reduces the accuracy of the mass estimation.

If we ignore the system error, (17) can be rewritten as,

$$F_{et} = m(gf + a_{sen}) \tag{20}$$

Consider this equation as the contrast model of mass estimation, which is the common form used by other scholars; the offline least squares algorithm is used, where $\varphi' = (gf + a_{sen})$ and $Z' = (F_{et})$ are the system input and output, respectively, and $\theta' = (m)^T$ is the parameter to identify.

Based on the principle of valid data, all valid data of each independent test were selected for the offline estimation. Two kinds of estimation models, (17) and (20), were used. Two typical cases of full-load situations are shown in Figures 9 and 10. The blue dots are valid data. Both real mass and two estimated masses are drawn as lines. The slope of each line represents different mass values.

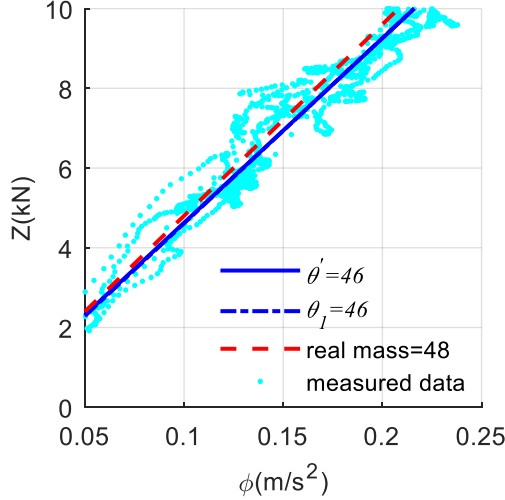

**Figure 9.** Total valid input and output of the offline estimation in one independent test (full load 1).

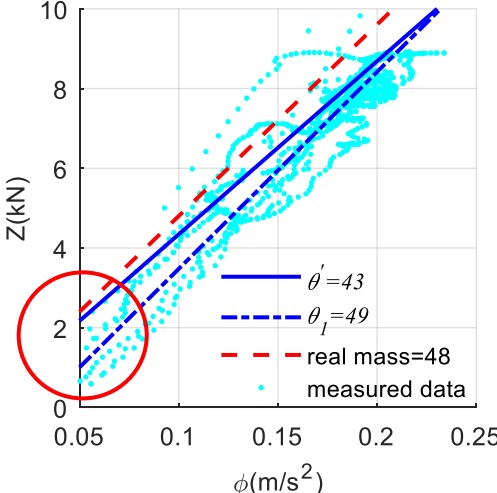

**Figure 10.** Total valid input and output of the offline estimation in one independent test (full load 2).

Figure 9 shows that there is only a slight difference between the two models when the system error is small. Its estimated system error is $\theta_2 = 428$ N, and the estimated mass is $\theta_1 = \theta' = 46$ t. Figure 10 shows that if the system error cannot be ignored, the accuracy is reduced with only one estimated parameter. The value of the system error is $-1462$ N; thus, the absolute value is large. When considering the system error, the estimated mass is 49 t. The mass value is 43 t without considering the system error, which is far from its real mass.

Rethinking this question by data fitting, there are two fitting parameters if we consider the system error: one linear coefficient and one constant value. The position of the system error in the coordinate space is unconstrained. With only one fitting parameter, the result must cross the original point. The residual error of the two parameters is clearly smaller than that with only one parameter.

Since the system error can be estimated from the model, it is helpful to quantitatively analyze the system error of longitudinal dynamics. The statistical analysis of the system error is illustrated in Table 6. The percentage of system error in each independent test is calculated as (21).

$$\varepsilon_i = \frac{F_{se}}{\overline{F}_{et}} \times 100\% \tag{21}$$

where $\varepsilon_i$ is the percentage of system error in test $i$. Because $F_{et}$ of each test is a variable, we calculate its mean value. $\overline{F}_{et}$ is the mean equivalent tractive force of test; $\varepsilon_i$ is the percentage of system error of each test.

**Table 6.** Statistical analysis of the system error.

| Tests Condition | Average (N) | Standard Deviation | Average of Absolute (N) | Average Percentage (%) | Average of Absolute Percentage (%) |
|---|---|---|---|---|---|
| full load trailer | −798.8 | 869.7 | 919.8 | −9.9 | 10.5 |
| empty trailer | 240.3 | 1158.5 | 872.5 | 8 | 16.8 |

The value of system error is random; its (absolute) average and standard deviation are large. For the mean system error, the full load is negative, whereas the empty trailer is positive. The absolute average percentages of system error are 10.5% and 16.8% for two loads, which indicates that the measured variables and parameters have unneglectable error. The system error has no linear relationship system input or output. The system error will inevitably introduce error when this constant value is fitted into the linear coefficient, which represents the vehicle mass.

### 5.3. Comparison of the Consideration of System Error

Although the measured data are not sufficiently qualified, it is convenient to achieve an acceptable mass result by modifying the model structure. (17) and (20) were used to estimated vehicle mass. The results are shown in Table 7.

**Table 7.** Statistical analyses of the estimated mass of the two models

| Tests Condition | | Estimated Mass (t) | Error (t) | Average Absolute Error (t) | Maximum Absolute Error (t) | Average Absolute Error Percentage in Real Load (%) |
|---|---|---|---|---|---|---|
| Full-load trailer | $\theta' = [m]^{\mathrm{T}}$ | 44.8 | −3.2 | 4.5 | 14.6 | **9.4** |
| empty trailer | $\theta = [m\ F_e]^{\mathrm{T}}$ | 48.1 | 0.1 | 1.9 | 4.6 | **4** |
| without trailer | $\theta' = [m]^{\mathrm{T}}$ | 26.9 | 3.7 | 5.3 | 19.9 | **22.8** |
| full-load trailer | $\theta = [m\ F_e]^{\mathrm{T}}$ | 24.7 | 1.5 | 2.1 | 7.3 | **9.1** |
| empty trailer | $\theta' = [m]^{\mathrm{T}}$ | 8 | -1.5 | 1.5 | 3.5 | **15.8** |
| | $\theta = [m\ F_e]^{\mathrm{T}}$ | 9.8 | 0.3 | 0.8 | 1.9 | **8.4** |
| total | $\theta' = [m]^{\mathrm{T}}$ | - | -0.3 | 3.8 | 12.7 | **16** |
| | $\theta = [m\ F_e]^{\mathrm{T}}$ | - | 0.6 | 1.6 | 4.6 | **7.2** |

Without system error, only 57.1% of the estimated mass error can be constrained in 3 t. This number can be improved to 87.9% with system error. Without system error, the total average absolute error percentage in real load is 16%. When the system error is considered, this number can be reduced to 8.8%. Above all, the mass estimation with system error consideration in vehicle longitudinal dynamics performs much better than that without system error.

## 6. Conclusions

Vehicle longitudinal dynamics contains too many constants and variables, and it is difficult to adapt various driving situations. Normal state estimators consider system error as white noise, but many sources of error, such as the accuracy of measured parameters, environment and vehicle motion state, make the system error become colored noise.

This paper considers system error as a parameter to be estimated. The RLS with two unknown parameters was used to identify both vehicle mass and system error. The source of system error was analyzed in both qualitative and quantitative aspects. The road tests indicate that the percentage of mass error is 16%, and, if the system error is considered, the percentage of mass error is 7.2%. When the system error is considered, the precision of mass estimation improves by 8.8%. The proposed model can offer online mass estimation for intelligent vehicle, especially for heavy-duty vehicle (HDV). Vehicle mass is an important parameter for advanced control systems such as automated gear shift strategies and economic cruise control.

Future research may focus on modeling system error with other reasonable methods. Here, we consider the system error as a constant value that is a balance between accuracy and efficiency. If the mechanism of system error can be expressed by a more reasonable form without increasing calculations, progress can be made in mass estimation.

**Author Contributions:** N.L. and S.S. conceived the main concept of the control structure and developed the entire control system. N.L. carried out the research and analyzed the numerical data with guidance from C.Z. N.L. and S.S. collaborated to prepare the manuscript.

**Funding:** This research was funded by Natural Science Foundation of China (51805202).

**Conflicts of Interest:** The authors declare no conflict of interest.

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
