# Peer review of "The Method of Mass Estimation Considering System Error in Vehicle Longitudinal Dynamics"

_energies, doi:10.3390/en12010052_

Round 1
Reviewer 1 Report
p.p1 {margin: 0.0px 0.0px 0.0px 0.0px; font: 12.0px Helvetica} p.p2 {margin: 0.0px 0.0px 0.0px 0.0px; font: 12.0px Helvetica; min-height: 14.0px}This paper presents a very interesting dynamic analysis including mass estimation for large vehicles. Literature survey is good and builds a concise insight into the position of the work in the field. The hypothesis of error is explored and investigated in the main sections. Below are a few comments for the authors to consider.
- please include definitions for all parameters on their first appearance.
- there are 2 equation 1, in page 2 and page 3.
- please justify or provide citations for the numerical coefficients in equations 1 and 2 on page 3.
- for section 3, are there photographs of the real test system? Figure 3 is useful, but it alone doesn’t evidence the real test.
- what was the brand and part number of the MEMS accelerometer used? authors mention 1.6g, is that range enough? With a sample frequency of 20 Hz, was that enough? I am afraid a lot of data might not actually be measured due to the small acceleration and frequency range. You can see in figure 4, some amplitude is cut off. Also, frequency information is very pixelated. I think a larger amplitude range and much higher sampling frequency is needed for accuracy.
- Are there any comparison of the results here with other methods in the literature? Perhaps including a short comparison in the discussion part, in order to position the results with the wider field.
Author Response
On behalf of all the authors, I express great appreciation for your careful review and comments of this paper. It is my pleasure that you recognized our outcomes. The answers to your comments are followed in order. Main changes of manuscript are marked in yellow background.
Point 1: please include definitions for all parameters on their first appearance.
Response 1: Equations have been modified. Please reference to the manuscript.
Point 2: there are 2 equation 1, in page 2 and page 3.
Response 2: Equations have been modified. Please reference to the manuscript.
Point 3: please justify or provide citations for the numerical coefficients in equations 1 and 2 on page 3.
Response 3: We provided the value of parameters in Table 2.
Point 4: for section 3, are there photographs of the real test system? Figure 3 is useful, but it alone doesn’t evidence the real test.
Response 4: Here is one photograph of the test vehicle(the attatched PDF file). While the Installation position of sensor cannot be expressed as clearly as the previous one. Since we provided the exact value of vehicle parameters, I hope this can evidence the real test
Point 5: what was the brand and part number of the MEMS accelerometer used? authors mention 1.6g, is that range enough? With a sample frequency of 20 Hz, was that enough? I am afraid a lot of data might not actually be measured due to the small acceleration and frequency range. You can see in figure 4, some amplitude is cut off. Also, frequency information is very pixelated. I think a larger amplitude range and much higher sampling frequency is needed for accuracy.
Response 5: The brand and part number of the sensor is Continental BSZ04. For HDVs, vehicle mass is very large. The power performance of longitudinal dynamics is not very intense. The maximum longitudinal acceleration is not more than 2m/s2. The measurement range of sensor is sufficient. Higher sampling frequency can only be measured by test equipment, cannot be implemented on real HDVs. The Vehicle Processing Unit can only offer measured data with 20Hz.
Point 6: Are there any comparison of the results here with other methods in the literature? Perhaps including a short comparison in the discussion part, in order to position the results with the wider field.
Response 6: We don’t comparied with other methods for the current manuscript. More road tests should be carried out for further comparison. We do not have the test condition right now. This may carried out in the future.

Reviewer 2 Report
The paper describes the methods and results of mass estimation of heavy-duty vehicles. However, based on the review, there are some concerns and suggestion needed to addressed as the follows:
In the heavy-duty driving motion, the slip rate of vehicles and tires is a major factors on the system mass and tract forces. I did not see the paper include this type of considerations in the study. In addition, the road conditions may significantly impact on the rotating and liner friction of powertrain. I am not also seeing the manuscript make clear assumptions on the tests and results.
In some equations, such as equations (1)-(9), there are many variables or inputs/output involved in the dynamics that I am not aware of lots of constants. But the authors state there are too many constants are not uncertain. Please clarify those constants and also explain why those constant are not certain because of measures, changing, or some other effects?
The manuscript includes many symbols, but does not provide the physical meanings and values. I suggest that the manuscript may list a full nomenclature list to clarify those.
The study also state the system errors and un-system errors. Please clarify what components or parameters or variable are belong to each category.
There are some errors on sequence numbers of equation and reference number wrong in the manuscript.
Author Response
On behalf of all the authors, I express great appreciation for your careful review and comments of this paper. It is my pleasure that you recognized our outcomes. The answers to your comments are followed in order. Main changes of manuscript are marked in yellow background.
Point 1: In the heavy-duty driving motion, the slip rate of vehicles and tires is a major factors on the system mass and tract forces. I did not see the paper include this type of considerations in the study. In addition, the road conditions may significantly impact on the rotating and liner friction of powertrain. I am not also seeing the manuscript make clear assumptions on the tests and results.
Response 1: In section 4.1, we summarized three source of system error. Tires slip is one of it. In 4.1.3, we mentioned that ‘Lateral motion and tire slip are ignored’. Current method (without considering slip) is a balance between accuracy and efficiency. If the slip estimation can be expressed by a reasonable form without increasing calculations, progress can be made in the future.
Point 2: In some equations, such as equations (1)-(9), there are many variables or inputs/output involved in the dynamics that I am not aware of lots of constants. But the authors state there are too many constants are not uncertain. Please clarify those constants and also explain why those constant are not certain because of measures, changing, or some other effects?
Response 2: A new table of parameters has been added, Table 2. These constant parameters may small change with different driving cycle. In section 4.1.2, we explain why some constants may change with driving condition.
Point 3: The manuscript includes many symbols, but does not provide the physical meanings and values. I suggest that the manuscript may list a full nomenclature list to clarify those.
Response 3 : Equations have been modified. Please reference to the manuscript.
Point 4: The study also state the system errors and un-system errors. Please clarify what components or parameters or variable are belong to each category.
Response 4: The so-called system error represents accumulative errors caused by sensor drift or uncertain environment disturbance in vehicle longitudinal dynamics. It contains the error of both variables and parameters in the tractive force and driving resistance. System error is designed to adjust complex error source with a simplified form. It cannot specify a parameters or variable to a certain category.
Point 5: There are some errors on sequence numbers of equation and reference number wrong in the manuscript.
Response 5: Equations have been modified. Please reference to the manuscript.
Reviewer 3 Report
The work is interesting, it concerns an important topic and for the most part it is written correctly, but some issues require clarification and possible improvement.
General remark:
The English language of work should be improved in some places. Starting from the beginning, the Reviewer suggests modifying the work title, e.g. : The method of mass estimation considering system error in the vehicle longitudinal dynamics. The words "that considers" do not work in the title.
Other examples:
line 30 - there is " importance", it should be "important"
line 30 - Authors defined HDV as Heavy Duty VehicleS, so phrase "a HDV" is inconsistent.
line 87 - there is " as while noise", Reviewer supposes that is should be "as white noise".
line 88 - there is " independence test", Reviewer supposes that is should be "independent test".
line 103 - there is " fly wheel", it should be " flywheel"
Detailed remarks:
It is needed to add an information about area of application (HD Vehicles) of such a method in the abstract too.
lines 90 - 91 - formulas in the text do not look good, place them just like formulas are placed usually.
lines 98-98 - number of the formula is mistaken, so the same remark concern the rest of formulas.
line 105 - " Error! Reference source not found." - What is this??
lines 149-150 - Authors should correct this sentence - now it suggests that the trailer has an engine.
line 161 - "ignition" - it is unclear what do the Authors mean?
Figure 4-6 - Titles of the vertical axis are illegible.
Is it possible using the developed method to detect an abnormal increase in rolling resistance e.g. due to a drop of a tire pressure, puncture of a tire, damage of a tire, brakes or a bearing? It may be very interesting from a point of view of active safety reasons.
Statement about progress of the work should be added in conclusions section: Is this method accurate enough to be used as it is now for purposes of "gear shift strategies and economic cruise control" or its further development is needed?
Author Response
On behalf of all the authors, I express great appreciation for your careful review and comments of this paper. It is my pleasure that you recognized our outcomes. The answers to your comments are followed in order. Main changes of manuscript are marked in yellow background.
Point 1: The English language of work should be improved in some places. Starting from the beginning, the Reviewer suggests modifying the work title, e.g. : The method of mass estimation considering system error in the vehicle longitudinal dynamics. The words "that considers" do not work in the title.
Response 1: Thank you for your suggestion, the title has been modified.
Point 2: line 30 - there is " importance", it should be "important"
Response 2: modified
Point 3: line 30 - Authors defined HDV as Heavy Duty VehicleS, so phrase "a HDV" is inconsistent.
Response 3: modified
Point 4: line 87 - there is " as while noise", Reviewer supposes that is should be "as white noise".
Response 4: modified
Point 5: line 88 - there is " independence test", Reviewer supposes that is should be "independent test".
Response 5: modified
Point 6: line 103 - there is " fly wheel", it should be " flywheel"
Response 6: modified
Point 7: It is needed to add an information about area of application (HD Vehicles) of such a method in the abstract too.
Response 7: The application has been added in the abstract.
Point 8: lines 90 - 91 - formulas in the text do not look good, place them just like formulas are placed usually.
Response 8: All formulas in the text have been deleted, and defined by equations with numbers.
Point 9: lines 98-98 - number of the formula is mistaken, so the same remark concern the rest of formulas.
Response 9: Equations have been modified. Please reference to the manuscript.
Point 10: line 105 - " Error! Reference source not found." - What is this??
Response 10: modified
Point 11: lines 149-150 - Authors should correct this sentence - now it suggests that the trailer has an engine.
Response 11: modified
Point 12: line 161 - "ignition" - it is unclear what do the Authors mean?
Response 12: modified
Point 13: Figure 4-6 - Titles of the vertical axis are illegible.
Response 13: The meanings of the vertical axis are illustrated in text, line 174.
Point 14: Is it possible using the developed method to detect an abnormal increase in rolling resistance e.g. due to a drop of a tire pressure, puncture of a tire, damage of a tire, brakes or a bearing? It may be very interesting from a point of view of active safety reasons.
Response 14: State estimation technologies can be used to detect abnormal condition of tire. More measurement variables with high accuracy are needed. May be vehicle mass should be regard as a known parameter for rolling resistance estimation. Such interesting topic can be performed in the future.
Point 15: Statement about progress of the work should be added in conclusions section: Is this method accurate enough to be used as it is now for purposes of "gear shift strategies and economic cruise control" or its further development is needed?
Response 15: The conclusions have been modified. The accuracy of mass estimation still has room for improvement. Controlled Maximum absolute error within 1~ 2 tons is an ideal situation. If the mechanism of system error can be expressed by a more reasonable form without increasing calculations, progress can be made in mass estimation.
Reviewer 4 Report
It is very important to develop reliable vehicle mass estimation methods that can be used in real-time situations. The manuscript presents an estimation method of vehicle mass under assumption that the system error is coloured noise. The paper appears to be well-written and comprehensively referenced. I have a few minor comments on the manuscript.
Comments:
Line 52. “provide” -> “provided”
Line 87. “while” -> “white”
Line 92. Insert space between “Network” and the bracket.
Line 96. Probably it should be (3) instead of (4).
Line 100. Probably it should be (2) instead of (3).
Line 105. Delete “Error! Reference source not found.” Insert a reference if necessary.
Line 111. Insert space between “used” and the bracket.
Line 123. Insert space between “as” and the bracket.
Formula (9). What is gamma?
Line 124. Provide more information about lambda and L. Interpret lambda.
Line 127. The phrase “the data acquired when the algorithm began should be gradually forgotten” is not clear. It should be rephrased.
Line 147. “a expressway” -> “an expressway”.
Line 234. Insert space between “models” and the bracket.
Line 240. Insert space between “EVs” and the bracket.
Author Response
On behalf of all the authors, I express great appreciation for your careful review and comments of this paper. It is my pleasure that you recognized our outcomes. The answers to your comments are followed in order. Main changes of manuscript are marked in yellow background.
Point 1: Line 52. “provide” -> “provided”
Response 1: modified
Point 2: Line 87. “while” -> “white”
Response 2: modified
Point 3 : Line 92. Insert space between “Network” and the bracket.
Response 3: modified
Point 4: Line 96. Probably it should be (3) instead of (4).
Response 4: modified
Point 5: Line 100. Probably it should be (2) instead of (3).
Response 5: modified
Point 6: Line 105. Delete “Error! Reference source not found.” Insert a reference if necessary.
Response 6: modified
Point 7: Line 111. Insert space between “used” and the bracket.
Response 7: modified
Point 8: Line 123. Insert space between “as” and the bracket.
Response 8: modified
Point 9: Formula (9). What is gamma?
Response 9: It is just an intermediate variable of numerical calculation. It does not have a physics meaning. So the explanation has been ignored.
Point 10: Line 124. Provide more information about lambda and L. Interpret lambda.
Response 10: λ and L are both parameters in the RLS algorithm, which can be selected by tests data. Here we tried several different value and selected appropriate ones. The explanations of these two parameters are in line 128~line 131.
Point 11: Line 127. The phrase “the data acquired when the algorithm began should be gradually forgotten” is not clear. It should be rephrased.
Response 11: This vague sentence has been deleted.
Point 12: Line 147. “a expressway” -> “an expressway”.
Response 12: modified
Point 13: Line 234. Insert space between “models” and the bracket.
Response 13: modified
Point 14: Line 240. Insert space between “EVs” and the bracket.
Response 14: modified
Round 2
Reviewer 1 Report
I am satisfied with the authors response and updates.
Reviewer 2 Report
The authors have addressed all my concerns and questions effectively. I would recommend it for a publication in the journal.
Reviewer 3 Report
Authors have made significant effort in order to improve their paper. Any doubts have been clarified. Missing information has been added. The reviewer appreciates work of the Authors done in such a short period of time and recommends the revised manuscript to be published in Energies journal.
Reviewer 4 Report
The authors have made the necessary changes.